# Transfer learning of multicellular organization via single-cell and spatial transcriptomics

**Yecheng Tan** [ID][1,2☯], **Ai Wang** [ID][3☯], **Zezhou Wang**[1,4], **Wei Lin** [ID][1,5,6*], **Yan Yan**[3*], **Qing Nie**[7*], **Jifan Shi** [ID][1,5,6*]

**1** Research Institute of Intelligent Complex Systems, Fudan University, Shanghai, China, **2** Institute of Science and Technology for Brain-Inspired Intelligence, Fudan University, Shanghai, China, **3** Department of Cardiology, Zhongshan Hospital, Fudan University, Shanghai, China, **4** Shanghai Center for Mathematical Sciences, Fudan University, Shanghai, China, **5** State Key Laboratory of Medical Neurobiology and MOE Frontiers Center for Brain Science, Fudan University, Shanghai, China, **6** Shanghai Artificial Intelligence Laboratory, Shanghai, China, **7** Department of Mathematics, University of California Irvine, Irvine, California, United States of America

☯ These authors contributed equally to this work.

\* wlin@fudan.edu.cn (WL); yan.yan@zs-hospital.sh.cn (YY); qnie@uci.edu (QN); jfshi@fudan.edu.cn (JS)

**Data availability statement:** All relevant data are within the paper and its Supporting

## Abstract

Biological tissues exhibit complex gene expression and multicellular patterns that are valuable to dissect. Single-cell RNA sequencing (scRNA-seq) provides full coverages of genes, but lacks spatial information, whereas spatial transcriptomics (ST) measures spatial locations of individual or group of cells, with more restrictions on gene information. Here we show a transfer learning method named iSORT to decipher spatial organization of cells by integrating scRNA-seq and ST data. iSORT trains a neural network that maps gene expressions to spatial locations. iSORT can find spatial patterns at single-cell scale, identify spatial-organizing genes (SOGs) that drive the patterning, and infer pseudo-growth trajectories using a concept of SpaRNA velocity. Benchmarking on a range of biological systems, such as human cortex, mouse embryo, mouse brain, *Drosophila* embryo, and human developmental heart, demonstrates iSORT's accuracy and practicality in reconstructing multicellular organization. We further conducted scRNA-seq and ST sequencing from normal and atherosclerotic arteries, and the functional enrichment analysis shows that SOGs found by iSORT are strongly associated with vascular structural anomalies.

## Introduction

Single-cell RNA sequencing (scRNA-seq)[1] provides high-resolution and comprehensive transcriptomic profiles for all genes, allowing systematic analysis of cellular heterogeneity[2], cell differentiation[3–5], and disease mechanisms[6]. Computational analysis tasks of scRNA-seq data includes clustering[7], cell types annotation [8], differentially expressed gene identification [9], and inferring pseudo-time [10]. Because the measured tissues are dissociated

Information files. The original public data used in this paper can be accessed through the following links: (1) FISH data from the Berkeley Drosophila Transcription Network Project (BDTNP): https://shiny.mdc-berlin.de/DVEX/; (2) 10X Visium data of the human dorsolateral prefrontal cortex (DLPFC): http://spatial.libd.org/spatialLIBD/; (3) Adult mouse cortical cell datasets (GEO accession GSE71585): https://www.ncbi.nlm.nih.gov/geo/query/acc.cgi?acc=GSE71585; (4) Mouse embryo data: https://content.cruk.cam.ac.uk/jmlab/SpatialMouseAtlas2020/; (5) Seurat objects ST data (10X Genomics Visium) of mouse brain: https://satijalab.org/seurat/articles/spatial_vignette.html; (6) 10X Visium data of mouse brain: https://www.ebi.ac.uk/biostudies/arrayexpress/studies/E-MTAB-11114; (7) MERFISH data of mouse brain: https://portal.brain-map.org/atlases-and-data/bkp/abc-atlas; (8) MERFISH data of human MTG: https://doi.org/10.5061/dryad.x3ffbg7mw; (9) SMART-seq data of human MTG: https://portal.brain-map.org/atlases-and-data/rnaseq/human-mtg-smart-seq; (10) Single-cell RNA-seq and Spatial Transcriptomics data of developing human heart: https://data.mendeley.com/datasets/mbvhhf8m62/2; (11) Human artery data: https://doi.org/10.5281/zenodo.15132967. The iSORT algorithm is available on GitHub: https://github.com/xiaojierzi/iSORT.

**Funding:** This work is supported by National Key Research and Development Program of China [2022YFC2704604 to J.S.], National Natural Science Foundation of China [No. 12301620 to J.S., No. 11925103 to W.L., No. 82070463 to Y.Y.], Science and Technology Commission of Shanghai Municipality [No. 21DZ1201402 to W.L.], the AI for Science Foundation of Fudan University [No. FudanX24AI041], and the Shanghai Municipal Data Bureau special fund for urban digital transformation [No. 202401065]. The funders had no role in study design, data collection and analysis, decision to publish, or preparation of the manuscript.

**Competing interests:** The authors have declared that no competing interests exist.

during the sequencing process, the information on the spatial locations of individual cells are lost in the scRNA-seq data.

Spatial transcriptomics (ST) [11] can simultaneously capture information of both gene expressions and cell locations, providing a more desirable approach to study multicellular spatial organization. The two main types of ST technologies include the image-based and the sequencing-based. Image-based methods, such as MERFISH [12], seqFISH [13], and STARmap [14], can detect only hundreds to thousands of genes at cellular or sub-cellular resolution. Sequencing-based methods, such as 10X Visium [15] and Slide-seq [16,17], can provide whole transcriptomic sequencing, but only have a resolution at spot of group of cells instead of individual cells. Stereo-seq [18] can capture thousands of genes in nanoscale resolution.

To utilize the strength of each data types, several computational tools were introduced to integrate scRNA-seq and ST data. For spot deconvolution, SPOTlight [19] used the non-negative matrix factorization; Cell2location [20] built a hierarchical Bayesian framework; Tangram [21] and STEM [22] employed deep neural networks on discrete spots. For estimating the single-cell location, novoSpaRc [23] and spaOTsc [24] used the optimal transport method to predict a spatial probability distribution for each individual cell; CSOmap [25] estimated cell-cell affinity through ligand-receptor interactions; scSpace [26] used a multilayer perceptron to extract features through transfer component analysis; CeLEry [27] employed deep learning with enhancing ST by data augmentation; and CellTrek [28] used a random forest approach with extensively interpolated ST data. Comparisons of those methods were carried out recently [29,30].

One major challenge for studying the multicellular organization using both scRNA-seq and ST data is to identify key genes that drive the spatial patterning of cells. Spatially variable genes (SVGs) [31,32], which are genes with high spatial variability in expressions, may mark the spatial domains in gene expression pattern, however, they are not necessarily the genes that are responsible for the formation of the spatial patterns. Here we introduce a quantity named spatial-organizing genes (SOGs) based on the concept of dynamical causality [33,34]. In other words, SOGs are characterized as the genes whose change critically affect the spatial organization of tissues.

The other major challenge is to demonstrate the differentiation trajectory of tissues in the physical space. RNA velocity proposed in 2018 used information of spliced and unspliced mRNA to derive a vector field in gene expression space presenting the direction of differentiation [35,36]. Further generations include scTour [37], which infer RNA velocity only by spliced mRNA expressions. However, RNA velocity does not consider the cellular organization and ignores the practical growth of cells in the physical space. Here we propose a quantity named SpaRNA velocity which projects the RNA velocity onto the ST slice, indicating pseudo-growth trajectories that model how cells transition to their progenies in space.

In order to estimate these two quantities, we utilize the density ratio technique in transfer learning by integrating a large amount of scRNA-seq data with one or a few ST slices as references. In this **i**ntegrative method for **S**patial **O**rganization of cells using density **R**atio **T**ransfer (iSORT), a function using a neural network is constructed to connect the spatial organization of single cells and the gene expression. With this function, we can estimate the sensitivity of the spatial pattern with respect to individual genes as a measure to quantify SOG. Meanwhile, SpaRNA velocity is obtained by transferring RNA velocity to the physical space. To validate the effectiveness of iSORT, we collected benchmark datasets from human dorsolateral prefrontal cortex (DLPFC), mouse embryo, mouse brain and human middle temporal gyrus (MTG) to test its accuracy and robustness in spatial reconstruction. *Drosophila* embryo

dataset was used to show iSORT's ability to reveal specific patterns. We collected human arteries affected by atherosclerosis and conducted scRNA-seq and ST sequencing experiments to investigate the role of SOGs in diseases associated with changes in spatial structure. SpaRNA velocity was visualized using the DLPFC dataset, a human developmental heart dataset and a mouse embryo dataset to illustrate pseudo-growth trajectories.

## Results

### An overview of iSORT

iSORT is a transfer learning-based framework which constructs a neural network mapping gene expression to spatial coordinates and further analyzes the spatial organization (Fig 1a). One or several low-resolution ST slices are used as references in the training process. iSORT integrates scRNA-seq and ST data, which can be sampled from heterologous sources with different cell-type distributions. Estimation of the density ratio is the core technique, which is a specialized method used in transfer learning to address distributional discrepancies between domains. There are three major steps in the framework of iSORT. Specifically,

**Step 1: Preprocessing** (Fig 1a). The inputs for iSORT, matched scRNA-seq data and ST slices, are preprocessed normalization, log transformation, and gene selection sequentially.

**Step 2: iSORT training** (Fig 1b). Denote $x$ as the gene expression after the preprocessing. Distributions of $x$ in scRNA-seq and ST data are usually totally different. iSORT first employs a reference-based co-embedding approach [38] to obtain a feature vector $z$, which is marked as $z = h(x)$. Cells with similar features have close spatial coordinates $y$ in the physical space, i.e. $p_{st}(y|z) = p_{sc}(y|z)$, where $p_{st}$ and $p_{sc}$ are distributions of the scRNA-seq and ST data, respectively. Then, to transfer the spatial information from ST to the scRNA-seq data, we estimate the mapping $y = g(z)$ by minimizing a loss function $L(z, y; g)$, where the density ratio $w(z) = p_{sc}(z)/p_{st}(z)$ measures the differences between scRNA-seq and ST data and is used for integrating the two types of data during training. In summary, iSORT constructs a mapping $f = g \circ h$ by neural networks, which assigns coordinates $y = f(x)$ to each cell with gene expression $x$.

**Step 3: Downstream analysis** (Fig 1c). iSORT reconstructs the spatial organization of cells by mapping the gene expression $x$ of each cell in scRNA-seq data to spatial coordinates $y$ using the function $f$. Results are naturally in single-cell resolution, and massive low-cost scRNA-seq data can be reorganized by using only one or a few ST references. For the downstream analysis: (1) We reveal spatial patterns by analyzing the distribution and clustering of specific gene expressions within the spatial coordinates $y$. (2) We identify SOGs by computing the index $I_g = \|\partial_{x_g} f(x)\|$, which quantifies the influence of each gene on the spatial structuring of the cell populations. (3) We define a quantity SpaRNA velocity as $v_{st} = f(x + v_{RNA}) - f(x)$, where $v_{RNA}$ is the classical RNA velocity [37] in the gene expression space. By integrating RNA velocity into our spatial model, one can analyze cellular dynamics on a ST slice, illustrating pseudo-growth trajectories of cells and elucidating how cells might migrate and evolve in physical space.

### Benchmarking iSORT in reconstructing the spatial organization of single cells

We first evaluated the performance of iSORT on reconstructing the spatial organization of three benchmark datasets, including the human dorsolateral prefrontal cortex (DLPFC) data, the spatially resolved mouse embryo data, and the mouse brain data. To further extend the evaluation, we also applied iSORT to reconstruct the spatial organization of the mouse

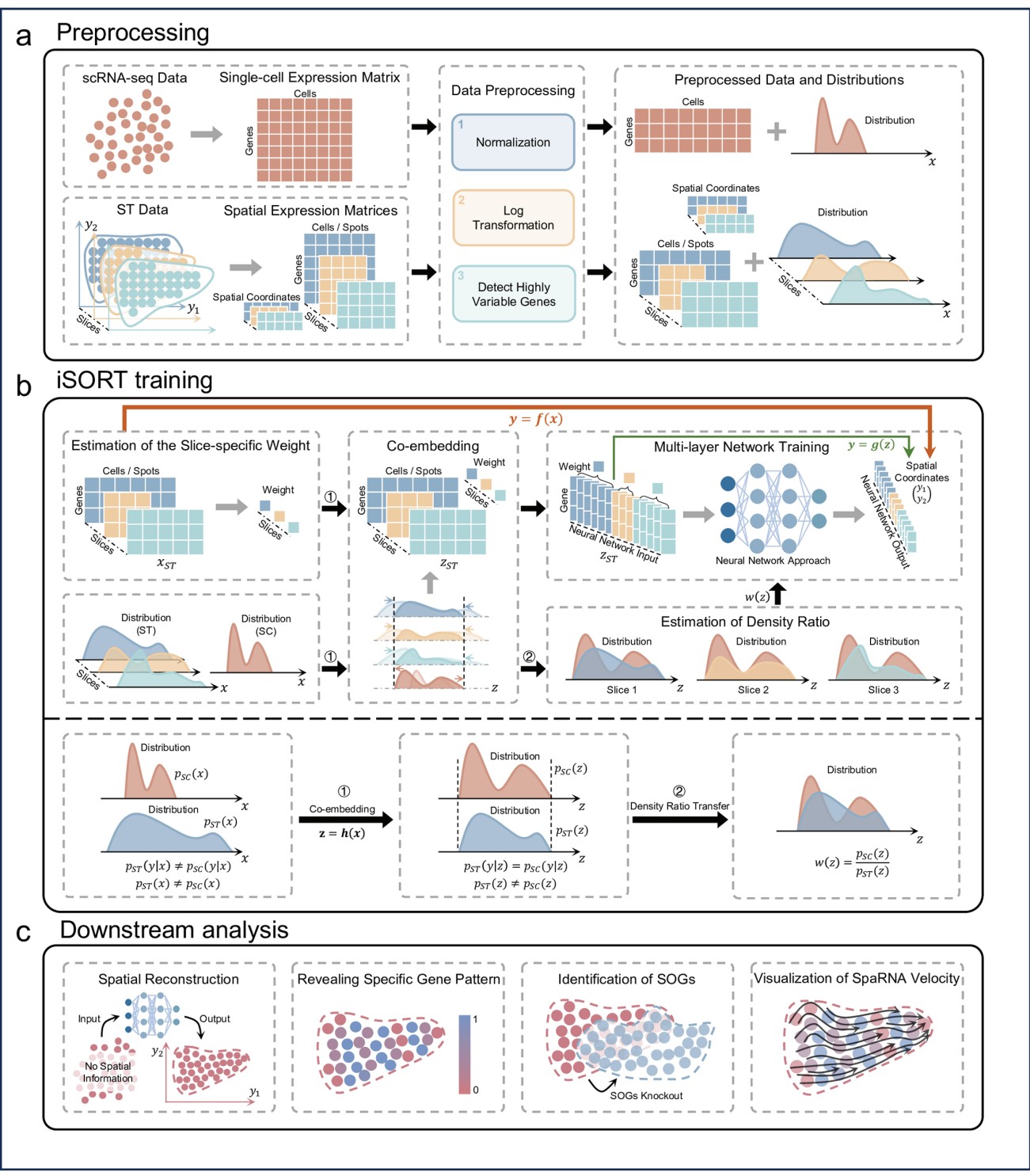

**Fig 1. Overview of iSORT.** (**a**) Preprocessing. Raw data from scRNA-seq dataset and ST slices are preprocessed and taken as iSORT's inputs. They then go through several pre-processing steps of normalization, log-transformation and selection of highly variable genes. Different samples could have diverse distributions of gene expressions. (**b**) Training. iSORT estimates the weights of each ST slice based on the scRNA-seq data and subsequently trains a mapping $f = g \circ h$ from gene expression $x$ to spatial location $y$, where $z = h(x)$ co-embeds data into the feature space in unified scale and $y = g(z)$ constructs a neural network combining slice-specific weights and density ratios. During training, each sample is estimated a weight based on the density ratio $w(z)$ of the scRNA-seq data to the ST data. (**c**) Downstream analysis. By the mapping $f$, iSORT can reconstruct spatial organization of tissues at single-cell resolution, reveal spatial expressive patterns of genes, identify SOGs, and visualize SpaRNA velocity in the physical space.

half-brain using 10X Visium and MERFISH, as well as the human middle temporal gyrus (MTG) region using MERFISH and Smart-seq. Details of the datasets are described in Note D in S1 Text.

**Reconstructing human DLPFC dataset.** The DLPFC data comprises three post-mortem brain samples, with each sample containing four ST slices [39]. The ST data were obtained by the spot-size resolution 10X Visium technology. We took the ST slice ID151674 as the scRNA-seq input by removing its spatial coordinates. The ST slice ID151675 was used as the reference. The output of iSORT were compared with the ground truth (ST slice ID151674) as well as five other existing methods capable of predicting single-cell spatial positions: scSpace [26], Tangram [21], novoSpaRc [23], CeLEry [27], and CellTrek [28]. Additionally, we performed a comparison with five spot deconvolution methods: RCTD [40], CARD [41], Redeconve [42], CytoSPACE [43], and Celloc [44], to provide a more comprehensive benchmarking analysis (Fig 2a). It is worth noting that Tangram is designed for spot deconvolution, while novoSpaRc is designed for imputing undetected genes, but both methods use probabilistic models and can be used for position reconstruction. Detailed information on the other methods is shown in Note B in S1 Text. In the group of methods for spatial position reconstruction, iSORT, Tangram, novoSpaRc, and CeLEry could reconstruct the shape of ST slice ID151674 from gene expressions, while iSORT, scSpace, novoSpaRc, and CeLEry could distinguish different layers with clear boundaries. For the spot deconvolution methods, we selected the cell type with the highest proportion in each spot as the dominant cell type for that spot, providing a unified basis for comparison. Each method was able to roughly distinguish the different layers, particularly the white matter (WM) layer. Among the spot deconvolution methods, Redeconve and RCTD showed the clearest distinction of layers. In contrast, CARD did not predict the presence of Layer 6 cell types, as no spot in the CARD predictions had Layer 6 as the most abundant cell type. Celloc and CytoSPACE showed distinct results for layers other than the WM layer, but the overall layer differentiation was less pronounced compared to Redeconve and RCTD. Four different indicators were used to evaluate the performance of the eleven methods (Fig 2b), i.e. the intra-layer similarity $S_C$, the normalized density distribution $\rho_C$, the aggregative volume index $A_C$, and the aggregative perimeter index $P_C$ (Definitions and details are provided in Note E in S1 Text). Values of the indicators measure the performance of algorithms in reconstructing layer structures. iSORT got an average $S_c$ as 0.94, $\rho_c$ as 0.84, $A_c$ as 0.88, and $P_c$ as 0.84, which were larger than the values obtained from the other ten methods (Figs 2b and A in S1 Text).

**Reconstructing spatially resolved mouse embryo dataset.** The mouse embryo data were sequenced by the seqFISH technology, involving sagittal sections at the 8-12 somite stage (E8.5-E8.75) [45]. The scRNA-seq input was taken as the gene expression from each spot by removing its spatial coordinates. As the spot of seqFISH reaches the single-cell resolution, in order to test the performance of iSORT on a low-resolution ST reference, we simulated a coarse-grained ST reference (details in Methods). The ground truth of the mouse embryo data is characterized by a rich diversity of cell types, which are displayed in a non-clustered pattern (Fig B in S1 Text). The spatial reconstructions of scRNA-seq data by scSpace and CeLEry mixed different cell types together and were generally unable to capture the inner structure of the mouse embryo (Fig B in S1 Text). novoSpaRc and Tangram were constrained by the sparsity of the simulated spots and cells were only assigned to the spots (Fig B in S1 Text). iSORT, however, reconstructed the spatial organization in a continuous space with single-cell resolution and retained the embryonic multicellular structure (Figs 2c and B in S1 Text).

**Reconstructing mouse brain dataset.** For the mouse brain datasets, the scRNA-seq and ST data were obtained from different samples, by different technologies, and with different cell-type distributions [46]. The scRNA-seq was conducted by Smart-seq2 and ST was sequenced

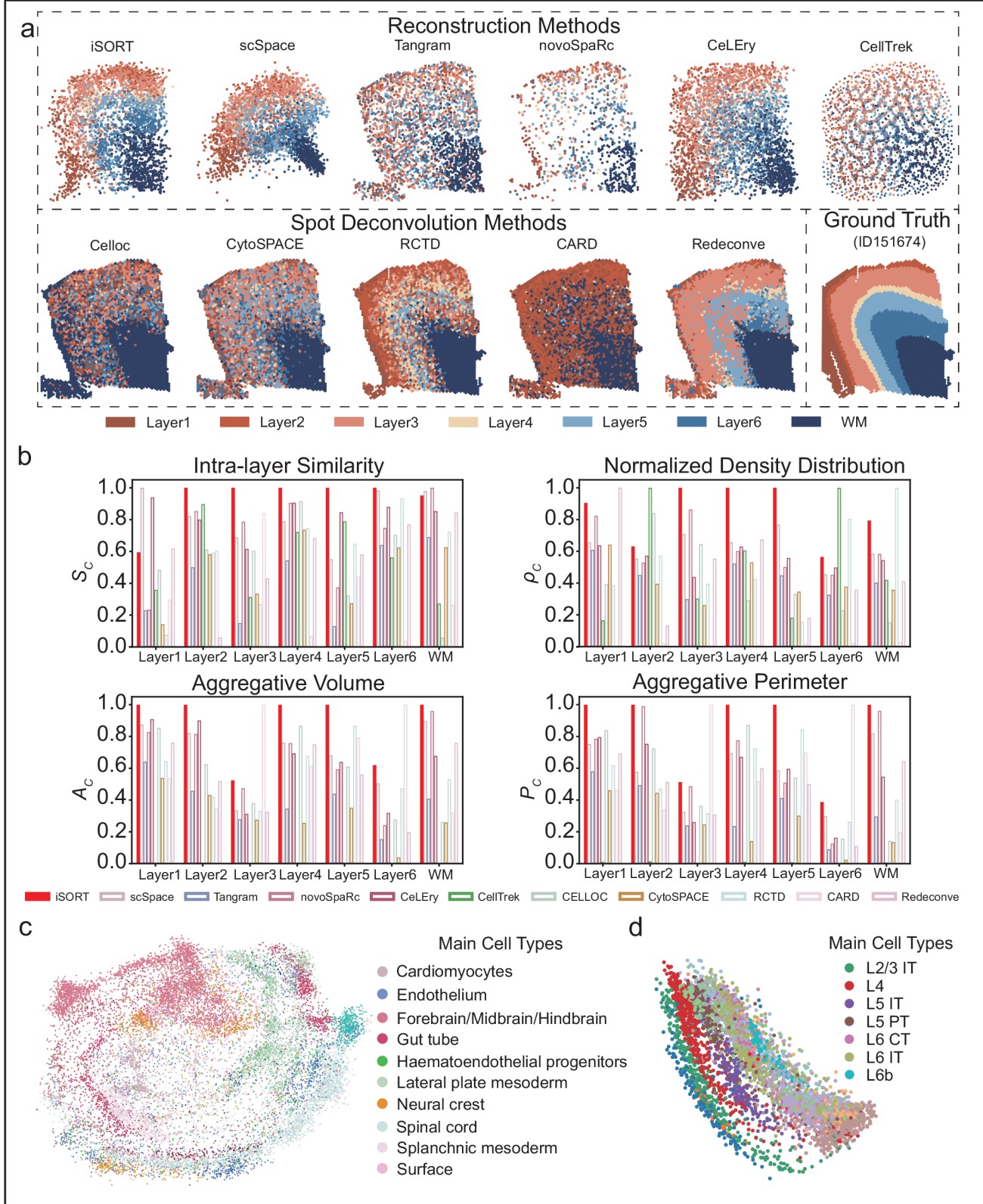

**Fig 2. Spatial organization reconstruction by iSORT on the DLPFC, spatially resolved mouse embryo, and mouse brain datasets.** (**a**) Ground truth and spatial reconstruction results for ID151674 in DLPFC dataset by eleven algorithms: iSORT, scSpace, Tangram, novoSpaRc, CeLEry, CellTrek, RCTD, CARD, Redeconve, CytoSPACE, and Celloc, where Tangram, RCTD, CARD, Redeconve, CytoSPACE, and Celloc are designed for spot deconvolution, whereas novoSpaRc is designed for imputing the undetected genes. Different colors represent different cortical regions. (**b**) Bar charts to compare

the performance of eleven algorithms based on four indicators: intra-layer similarity ($S_C$), normalized density distribution ($\rho_C$), aggregative volume index ($A_C$) and aggregative perimeter index ($P_C$) of each layer. The indicators of iSORT (the red filled bars) are above 0.875 in average, larger than the other ten algorithms. (**c**) Spatial reconstruction of the spatially resolved mouse embryo dataset with iSORT. Different colors represent different cell types. iSORT can reorganize single cells based on their gene expressions in a continuous space. (**d**) Spatial reconstruction of the mouse brain dataset with iSORT. iSORT can recover the stratified structures in the mouse brain.

by 10X Visium. Using iSORT, we reconstructed the spatial organization of cells in scRNA-seq data (Fig 2d). iSORT reconstructed the stratified architecture of the cerebral cortex, which reflected the sequential arrangement of seven laminar excitatory neuron subgroups: L2/3 intratelencephalic (IT), L4, L5 IT, L5 pyramidal tract neurons (PT), L6 IT, L6 corticothalamic (CT), and L6b. scSpace and CeLEry hardly present the stratified structures in the continuous space, while novoSpaRc and Tangram are limited by the spot resolution (Fig C in S1 Text).

*Reconstructing different brain regions with diverse technologies.* To further validate the performance of iSORT across different technical datasets, we designed two independent experiments. The first experiment aims to test the ability of iSORT to integrate sequencing-based and image-based technologies, particularly integrating MERFISH and 10X Visium technologies to reconstruct the spatial structure of the mouse half-brain. The 10X Visium dataset [20] contains 5 slices, with each slice consisting of several thousand samples and covering approximately 30,000 genes, making it suitable for analyzing cortical region and layer structures. We selected one slice (ST8059048) for analysis. The MERFISH [47] technology provides a brain atlas dataset, and we selected a region slice similar to the 10X Visium data (Fig Da in S1 Text), which contains over 100,000 cells and 550 genes, offering higher single-cell resolution and spatial coordinate information. By removing the coordinates from the 10X Visium data, we used the MERFISH slice as a reference to successfully perform spatial reconstruction of the 10X data. The results show that iSORT was able to clearly reconstruct the spatial structure of key brain regions, including multiple cortical layers, hippocampal regions, hypothalamus, and thalamus (Fig Db in S1 Text). The second experiment aims to test iSORT's ability to integrate image-based slices with single-cell data. We combined two human MTG datasets, one based on MERFISH [48,49] and the other based on Smart-seq sequencing [50]. Although the Smart-seq data lacks spatial coordinates for the cells, it includes anatomical information about brain subregions and manually annotated cell types. In contrast, the MERFISH dataset provides single-cell resolution data with spatial coordinates and layer information. In this analysis, iSORT successfully reconstructed the spatial organization of the human MTG, accurately distinguishing cortical layers (L1–L6) and restoring their layered structure (Fig E in S1 Text).

## Robustness of iSORT in spatial reconstruction across different slices and noise conditions

The ST slices can be sampled from heterologous data sources, with diverse cell-type distributions, and being sequenced under different structural distortions. These variations pose challenges to the robustness of spatial reconstruction methods. Additionally, in real-world scenarios, scRNA-seq data often contain varying levels of noise, which may further impact reconstruction accuracy. To systematically evaluate the robustness of iSORT under these conditions, we conducted two types of sensitivity analyses. First, we tested how iSORT performs when using different ST slices as references. We tested the human DLPFC dataset and used the same 'scRNA-seq' data (ST ID151674 with spatial information removed) to reconstruct its spatial structure. The original ST slices of DLPFC from different samples exhibited various

hierarchical distributions (Fig F in S1 Text). We systematically analyzed six different cases, each using different ST references (Fig 3a). Second, to assess the impact of noise on spatial reconstruction, we performed a noise sensitivity analysis on the 'scRNA-seq' data from ST slice ID151674. By introducing noise at different intensities, we simulated increasing levels of measurement variability.

**Case I: Single homologous ST reference.** The single ST reference ID151675 was from the same DLPFC region as the scRNA-seq (ID151674). The reconstruction of iSORT (Fig 3aI) is evaluated by four clustering indicators (Fig 2b). We also measured the accuracy of the reconstruction, as defined in Note E in S1 Text, which reaches 100% for an exact spatial reconstruction but 0% for a random guess. iSORT obtains an accuracy of 74.1% for the case with a single homologous ST reference (Fig 2d and Table A in S1 Text).

**Case II: Single heterologous ST reference.** The single ST reference ID151671 was from a heterologous tissue. iSORT obtained a spatial reconstruction with an accuracy of 75.4% (Fig 3aII and 3d). Another heterologous reconstruction using ID151507 (Case II') as the reference got an accuracy 68.0% (Fig G in S1 Text). In Case II and II', structural layers of the target sample were recovered as well as Case I.

**Case III: Multiple homologous ST references.** In this case, we used three homologous slices (ID151673, ID151675, ID151676) as the ST references. iSORT achieved a higher accuracy of 86.2% in reconstructing spatial organization than Case I, which is the highest value among all six cases (Fig 3aIII and 3d and Table A in S1 Text). The results implied that using multiple ST references could improve the accuracy of iSORT.

**Case IV: Multiple heterologous ST references.** In practice, scRNA-seq data and ST slices usually originate from different samples with different shapes and cell-type distributions. In this case, we used slices ID151507, ID151675, and ID151671 as the ST references. Although the diversity and complexity between the scRNA-seq and ST data were high, iSORT still reconstructed the hierarchical structure with an accuracy of 82.6% (Fig 3aIV). For other combinations of heterologous references, we present Case IV' using slices ID151675, ID151507, and ID151508 with an accuracy of 79.7%, and Case IV" using slices ID151675, ID151670, and ID151671 with an accuracy 81.5% (Fig G in S1 Text).

**Case V: Multiple homologous ST references with distortion.** To further test the robustness of iSORT, we added rotations to the ST references, which is one of the most common batch effects of experimental data. Homologous ST slices ID151673, ID151675, and ID151676 as in Case III were rotated by 45, 0, and −45 degree, respectively (Fig H in S1 Text). The accuracy value 82.6% (Fig 3aV) showed that iSORT could produce spatial reconstruction even when there were noticeable distortions in the input slices.

**Case VI: Multiple heterogeneous ST references with distortion.** The heterogeneous ST slices ID151507, ID151675, and ID151671 as in Case IV were rotated by 30, 0, and −40 degree, respectively (Fig H in S1 Text). iSORT's reconstruction captures the spatial organization with an accuracy of 79.1% (Fig 3a).

We also studied the spatial distributions of the true cell positions and reconstructions in different cases by violin plots (Fig 3b and 3c). An index of accuracy (Note E in S1 Text) was computed for comparing different cases, where Case III held the largest similarity (highest accuracy) with the ground truth (Fig 3d). iSORT obtained an accuracy over 74.0% in most cases (except Case II' with 68.0%), which quantitatively demonstrated that iSORT was able to robustly integrate the DLPFC spatial information from multiple ST slices under various conditions (Fig 3d and Table A in S1 Text).

**Noise sensitivity analysis for iSORT on DLPFC dataset.** To assess the robustness of iSORT under varying noise conditions, we performed a noise sensitivity analysis on the semi-simulated scRNA-seq data from ST slice ID151674. Gaussian noise was added to the data

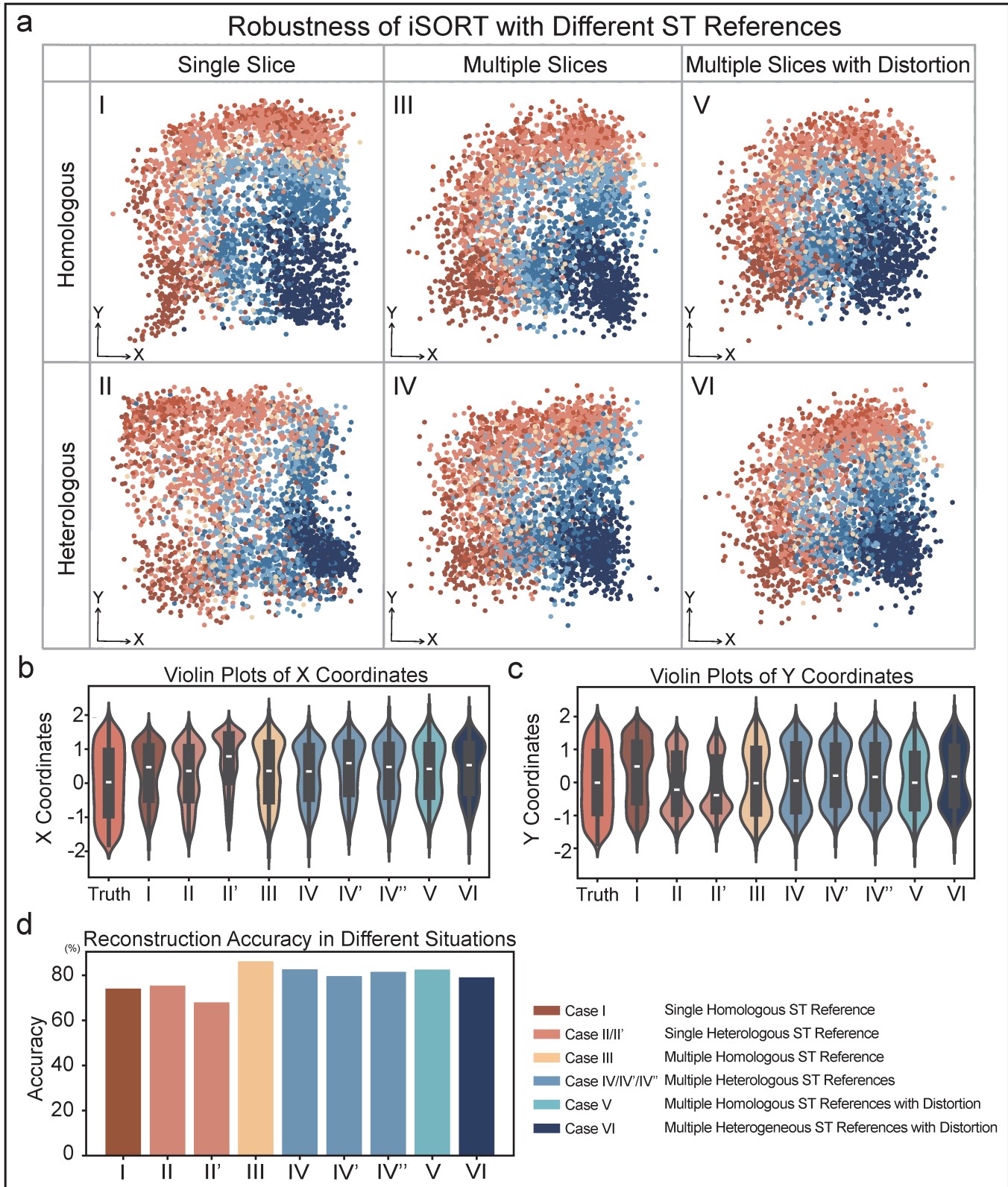

**Fig 3. Robustness of iSORT in reconstructing human DLPFC slice ID151674 with different ST references.** (a) iSORT's results with different ST references. Case I: Reconstruction using one homologous slice (Sample ID151675). Case II: Reconstruction using one heterologous slice (Sample ID151671). Case III:

Reconstruction using three homologous slices (Sample ID151673, ID151675, and ID151676). Case IV: Reconstruction using three heterologous slices (Sample ID 151675, ID151607, and ID151671). Case V: Reconstruction using three rotated homologous slices (Sample ID151673, ID151675, and ID151676). Case VI: Reconstruction using three rotated heterologous slices (Sample ID151675, ID151671, and ID151507). (**b**) Violin plots of X coordinates across different reconstruction scenarios, depicting the distributions of cells on the X-axis. Case II': Reconstruction using one heterologous slice (Sample ID151570). (**c**) Violin plots of Y coordinates across different reconstruction scenarios, depicting the distributions of cells on the Y-axis. Case IV': Reconstruction using three heterologous slices (Sample ID151675, ID151507, and ID151508). Case IV'': Reconstruction using three heterologous slices (Sample ID151675, ID151670, and ID151671) (**d**) The accuracy of iSORT across different cases. Reconstruction using a single heterogeneous slice will be slightly less effective compared to homogeneous slices, but if multiple heterogeneous slices are used, the reconstruction results are better than single slice in the sense of reconstruction accuracy.

at varying levels, with the noise intensity controlled by the parameter $\sigma$ (details in Methods). We tested noise levels of $\sigma = 0.1, 0.5, 0.8, 1, 2, 3$, simulating increasing levels of noise. The results showed that iSORT performed well within the range of $\sigma = 0$ to $1$, successfully reconstructing the spatial organization of the DLPFC tissue (Fig Ia in S1 Text). However, when $\sigma$ increased to 2, some performance metrics, such as intra-layer similarity $S_C$ and the aggregative volume index $A_C$, dropped below 0.3 on average (Figs Ib and J in S1 Text). At $\sigma = 3$, the decline became more pronounced, with certain values falling below 0.1, indicating a significant decrease in reconstruction accuracy. Despite this decline, iSORT preserved some spatial structure, indicating partial robustness under high noise levels.

## Uncovering spatial strip patterns of gene expression in *Drosophila* embryo

To study how iSORT uncovers the spatial organization of gene expression pattern, we applied iSORT to the *Drosophila* embryonic development dataset [51,52]. The ST data was sequenced by FISH-seq. The scRNA-seq input was obtained by removing the spatial information, while a low-resolution ST reference was obtained by the coarse-grained simulation (Fig 4b and more details in Methods). During the early development of embryo, several genes, such as *ftz*, exhibit unique spatial patterns and play decisive roles in the axial establishment and segmentation of the *Drosophila* body [53]. As an example, *ftz* in the original ST slice showed a unique seven-striped spatial pattern (Fig 4a). iSORT's reconstruction showed the seven stripes, each corresponding to a future body segment (Fig 4c). The true location distributions of single cells with the predicted ones by iSORT were compared, and iSORT captured the cell density of the embryo (Fig 4e). The marginal densities and corresponding errors between the true *ftz* expression and the predicted ones by iSORT were calculated, with reconstructed marginal densities on the x-axis showing the seven-stripe pattern of *ftz* (Fig 4f). We also compared the reconstructions with those generated by scSpace, Tangram, novoSpaRc, and CeLEry (Fig 4d). Tangram and novoSpaRc failed to fully recover the seven-stripe pattern, due to the discrete low-resolution constraints. scSpace and CeLEry could not clearly separate different stripes or determine the stripe number. Furthermore, we tested iSORT on 12 other genes, including *Antp, cad, Dfd, eve, hb, kni, Kr, numb, sna, tll, twi*, and *zen*. The coarse-grained references (Fig L in S1 Text) could not reflect the true spatial patterns of these 12 genes (Fig K in S1 Text), while the reconstructions by iSORT can reveal the corresponding patterns (Fig M in S1 Text). These results supported that iSORT could reveal spatial patterns of gene expression with good consistency with the original spatial organization.

## SOG analysis and identification of atherosclerosis-related biomarkers in human artery experiments

SOGs offer an approach to identify key genes determining the cellular spatial organization within tissues. Since iSORT provides a mapping $y = f(x)$ from the gene expression $x$ to the spatial coordinates $y$, the SOG index $I_g = \|\partial_{x_g} f(x)\|$ can measure the influence of gene $g$ (see

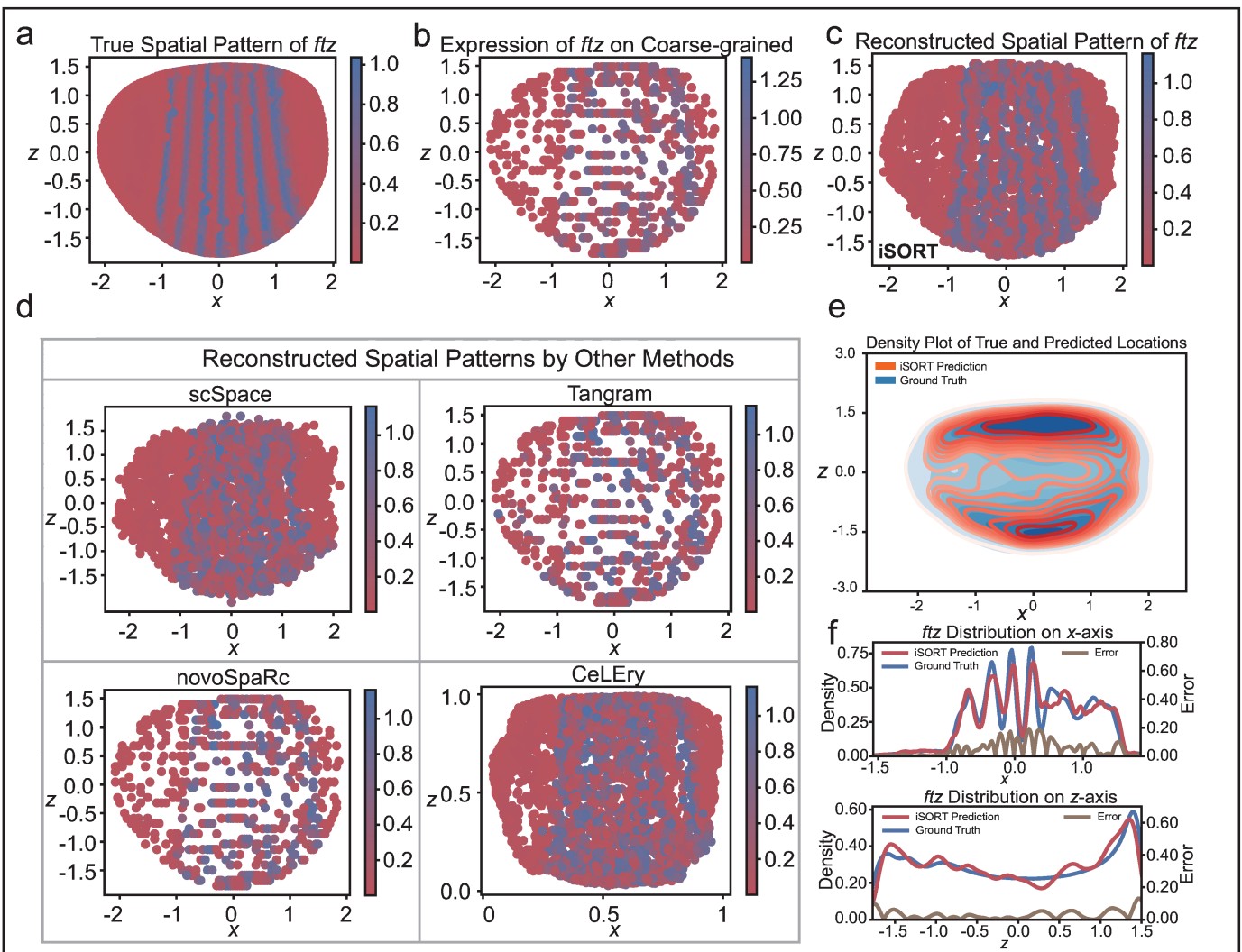

**Fig 4. Revealing the spatial pattern of the *ftz* gene from *Drosophila* embryo data using iSORT.** (**a**) Visualization of *ftz*'s spatial pattern in the original ST slice. (**b**) Visualization of *ftz*'s expression in the simulated coarse-grained ST reference, demonstrating the disruption of gene patterns in the low-resolution ST spots. (**c**) Visualization of *ftz*'s spatial pattern reconstructed from the scRNA-seq data by iSORT. Despite the fact that reference has lost the seven-stripe pattern, iSORT successfully restored the spatial distribution of the *ftz* seven stripes. (**d**) Reconstructed spatial patterns by scSpace, Tangram, novoSpaRc, and CeLEry. (**e**) Density plot contrasting the true (blue) and predicted (red) spatial locations of cells. The spatial density distribution of the iSORT reconstruction results is consistent with the ground truth. (**f**) Marginal densities for the true (blue) and predicted (red) spatial distributions of the *ftz* gene. The errors between the true and predicted densities are shown by the yellow line. The seven stripes along the x-axis are recovered by iSORT.

Methods). Genes with larger $I_g$s has more impact on the spatial organization. The top genes with large $I_g$ values are defined as SOGs. To study SOGs, we analyzed a simulated case and two different datasets.

**SOGs in a simulated experiment.** To distinguish between SOGs and SVGs, we used a ST toy model, as described in the Methods section. In this model, we generated a gene expression matrix and spatial coordinates, incorporating spatial dependencies to simulate realistic gene distributions. The first four genes were used to determine the spatial coordinates, while the remaining six genes were generated based on spatial weights with high spatial autocorrelation. Based on Moran's I index, the spatial autocorrelation of the first four genes was relatively low,

placing them behind the other six genes in terms of spatial variability (Table C in S1 Text). In contrast, the remaining six genes exhibited high spatial autocorrelation, which was visually apparent as a spatial clustering effect (Fig N in S1 Text). As a result, these six genes were identified as SVGs. However, in this model, the spatial positions were solely determined by the first four genes, as they directly influenced the tissue's spatial organization. After training iSORT, we performed SOG scoring, where iSORT successfully identified the first four genes as SOGs, ranking them as the top genes (Table B in S1 Text). This result highlights that iSORT recognizes the significant influence of these genes on the spatial structure, despite their lower spatial autocorrelation compared to the other genes. This experiment demonstrates the ability of iSORT to accurately distinguish SOGs from SVGs, providing insights into the genetic determinants of spatial organization.

***SOGs in human DLPFC dataset and in-silico knockout validations.*** We chose ID151674 (removing spatial information) as the scRNA-seq input, and ID151675 was used as the ST reference. With all 3635 genes, iSORT's reconstruction distinguished the hierarchical structure of the different cerebral cortexes (Fig 5a). After ranking the genes by the SOG index $I_g$, we conducted the knockout validation. When the top 20 SOGs (Table D in S1 Text) were removed, the reconstructed spatial organization was significantly altered (Fig 5b). When the top 300 SOGs were removed, the spatial structure was further affected (Fig 5c). We calculated the mean square error (MSE) between the true cell locations and the reconstructed ones. MSE value was found to increase with the number of knockout SOGs (Fig 5d). Moreover, we conducted the top-20-knockout experiments for SVGs selected by Moran's $I$ score [54] and SpatialDE [31] (Tables E and F in S1 Text). Moran's $I$ score assessed the spatial correlation and SpatialDE leveraged a Gaussian-process-based model on spatial expression (details in Note C in S1 Text). It was found that the spatial structures after knocking out SVGs were not significantly disrupted (Fig 5e and 5f) compared to the SOGs (Fig 5b). The results indicated SOGs selected by iSORT instead of SVGs were the key genes to maintain tissue's spatial organization. Additionally, we visualized the expression patterns of the top 20 SOGs in both the original ST and the reconstructed space (Figs O and P in S1 Text). These visualizations show how the expression of these key genes correlates with the spatial structure, further supporting their critical role in maintaining the tissue's spatial organization.

***Human artery sequencing and analysis of atherosclerosis-related SOGs.*** Diseases like atherosclerosis (AS) may involve changes in spatial morphological features like subendothelial lipid deposition, narrowing of the vascular lumen, and thickening of the arterial wall [55]. Identification of disease-related SOGs can provide instructive information for the treatment [56]. To validate the potential of SOGs, we sequenced a new dataset from human diseased arteries with AS and normal arteries. We performed scRNA-seq on Illumina NovaSeq platform and obtained ST by 10X Genomics. We also conducted hematoxylin and eosin staining to show that diseased arteries appear to be blocked while the normal artery was in a circular shape (Fig 5hI and 5hII). The detailed information of data collection and sequencing can be found in Methods section.

Because the artery is not a simple connected region but a circular shape (Fig 5g), we applied a reversible polar transformation to the ST data in preprocessing before using iSORT to reconstruct the spatial organization (Note F in S1 Text). There were two independent scRNA-seq samples as inputs: one from an individual without AS, and the other from a patient with AS. Three ST slices were used as multiple heterologous references: two from individuals without AS, and one from a patient with AS. The scRNA-seq and ST slices were sampled from totally different sources. The iSORT reconstructions for the normal artery and the diseased artery with AS showed the correct sequence of layers from inside to outside: endothelial cells (ECs), smooth muscle cells (SMCs), and fibroblasts (Fig 5hIII and 5hIV).

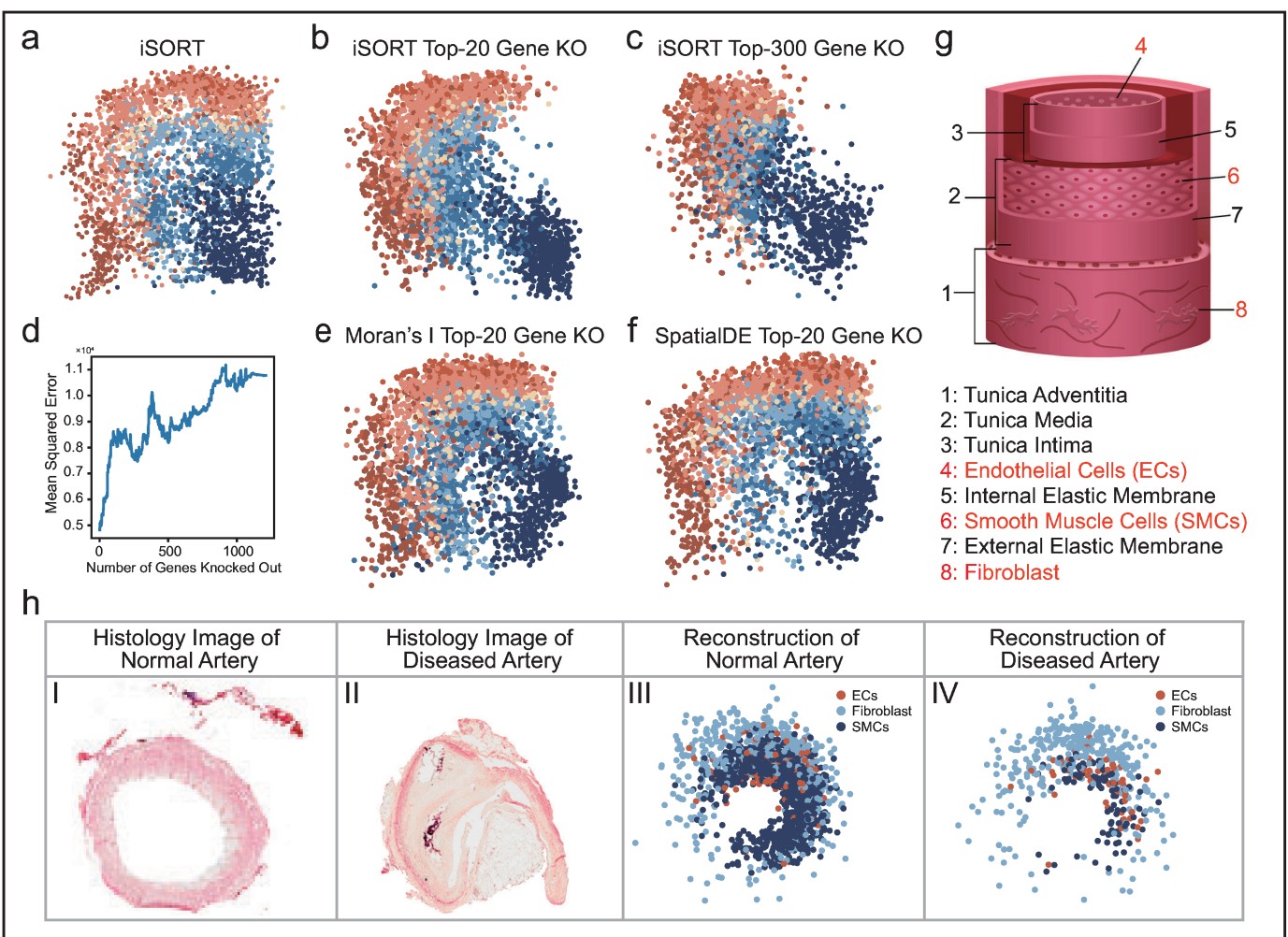

**Fig 5. In-silico gene knockout experiments on DLPFC dataset and analysis of the human artery dataset.** (**a**) Reconstruction of DLPFC by iSORT without gene knockout. (**b**) Reconstruction of DLPFC with the top-20 SOGs knocked out. Knocking out the first 20 SOGs disrupts the structure of the cerebral cortex. (**c**) Reconstruction of DLPFC with the top-300 SOGs knocked out. The structural disruption in the cerebral cortex is intensified, with increased mixing of cells across cortical layers. (**d**) Curve of the mean squared error (MSE) in reconstruction with increasing SOGs knocked out. The more genes that are knocked out, the worse the reconstruction is in the sense of MSE. (**e**) Reconstruction of DLPFC with the top-20 Moran's $I$ SVGs knocked out. (**f**) Reconstruction of DLPFC with the top-20 SpatialDE SVGs knocked out. (**g**) Schematic diagram of artery structure illustrating layered composition: the innermost layer is lined with endothelial cells (ECs), followed by smooth muscle cells (SMCs), and the outermost layer composed of fibroblasts. (**h**) Hematoxylin and Eosin staining and the reconstruction results of the normal artery and the diseased artery with atherosclerosis (AS): Panel I: Histology image of a normal artery. Panel II: Histology image of an artery with AS. Panel III: Reconstruction result for a normal artery, showcasing ECs, SMCs, and fibroblasts. Panel IV: Reconstruction result for a diseased artery with AS, showcasing ECs, SMCs, and fibroblasts. The reconstruction results by iSORT distinguish the hierarchical structure of the three cell types.

More detailed reconstructions for all cell types are exhibited in Figs Q and R in S1 Text. Compared with the normal artery (Fig 5hIII), the spatial organization of the diseased artery with AS was more concentrated on one side of the vessel (Fig 5hIV). Accumulation of lipids and fibrosis within the vessel wall in patients with AS, makes more cells concentrate on one side and lead to hardening and narrowing of the artery [55]. Then, we ranked and selected top SOGs of normal and diseased arteries based on $I_g$ (Fig S and Tables G and H in S1 Text). We found 16 overlapping SOGs which are not only closely associated with vascular function but also regulate vascular spatial structure. Among them, *TNN* achieved the highest $I_g$

score, which plays a crucial role in facilitating integrin binding and is essential for cell adhesion, migration, and proliferation [57]. *TNN* is also important in neuronal generation and osteoblast differentiation, further underscoring its significance in vascular structure and function [58,59]. Meanwhile, a number of genes found by $I_g$, such as *RFLNA*, *SPN*, and *EZH2*, are proved to be associated with AS [60–62]. We further performed the GO enrichment analysis on the top-50 SOGs. The GO analysis revealed that 11 out of the top 20 results in GO Biological Processes (BP) were common to both the normal and AS samples (Tables I and J and Figs T and U in S1 Text). The shared BP terms indicated commonality in core biological functions, while the unique BP terms appeared in the AS sample were highly related to the abnormal vascular mineralization during AS process. More detailed analysis of the AS related pathology can be found in Note H in S1 Text. These results indicate that the SOGs identified by iSORT contributed to maintaining vascular function and could serve as biomarkers for the study of AS. Further gene and functional analysis suggests that these SOGs regulate vascular spatial structure and are involved in the pathological processes of the vessel, underscoring their important role in spatial tissue reconstruction.

## SpaRNA velocity and pseudo-growth trajectories on human DLPFC, developmental heart and mouse embryo datasets

Next we study the iSORT derived SpaRNA velocity and its effect on the tissue organization using three datasets.

*SpaRNA velocity on human DLPFC dataset.* ST slice ID151674 was used in this experiment, with UMAP showing the distribution of cells (Fig 6a). It was known that the growth starts from the white matter (WM) to the Layer 1, passing through Layer 6, Layer 5, Layer 4, Layer 3 and Layer 2 [39,63,64]. Since scTour [37] is a method that infers pseudo-time and RNA velocity based solely on gene expression, which is not spatially resolved, its results are typically visualized in reduced-dimensional space. In this case, the pseudo-time and RNA velocity inferred by scTour in the reduced two-dimensional space could not capture the correct growth of Layer 1 (Fig 6b and 6c). Pseudo-time inferred from the gene expression space exhibited discontinuities on the ST slice (Fig 6d), such as between Layer 2 and Layer 3 (Fig 6e) [64]. However, the SpaRNA velocity derived from iSORT reconstructed the correct growth trajectories from WM to the sequential layers. Moreover, iSORT also allows the visualization of the detailed internal growth within a layer, particularly in the WM layer (Fig 6f).

*SpaRNA velocity on human developmental heart dataset.* Next, we presented the SpaRNA velocity derived by iSORT on the human developmental heart dataset [65]. This dataset contained ST slices from three developmental stages at 4.5-5, 6.5, and 9 post-conception weeks (PCWs). ST in the spot-size resolution with annotated cell types at 9 PCW showed the structure of a heart (Fig 6g). The initial development involves ventricular myocardial cells, preceding cells located in the atrial region, including atrial cardiomyocytes, epicardium-derived cells, smooth muscle cells, and fibroblasts, which is in accordance with the SpaRNA velocity inferred by iSORT (Fig 6h). Results for cells at 4.5-5 PCW and 6.5 PCW also showed similar pseudo-growth trajectories (Fig V in S1 Text). The chronological progression of heart development in physical space, in line with established biological processes [66,67], can be quantitatively described by the pseudo-growth trajectories obtained from SpaRNA velocity.

*SpaRNA velocity on mouse embryo dataset.* To further assess iSORT's ability to infer spatial RNA velocity, we tested its performance on a more complex mouse embryo tissue dataset, which was previously used for spatial reconstruction in our study [45]. The data contains rich cellular diversity across spatially resolved samples, which provided an excellent basis for

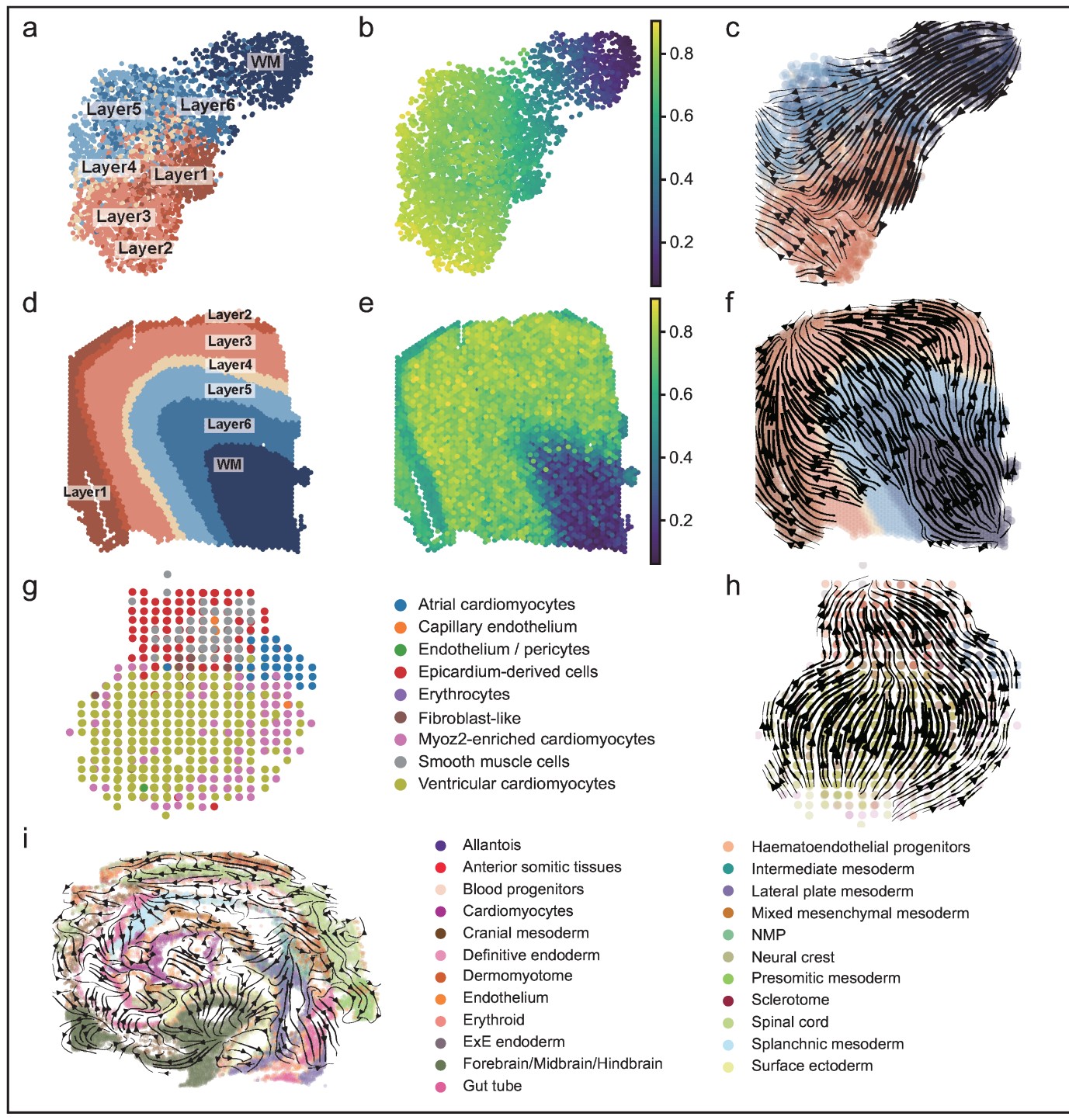

**Fig 6. Visualization of SpaRNA velocity on the DLPFC, human developing heart and mouse embryo datasets.** (**a**) Sample ID151674 shown in UMAP-reduction gene expression space annotated by different layers. In the low-dimensional gene space, the hierarchical organization of the cerebral cortex is arranged differently from the normal sequence. (**b**) Pseudo-time of sample ID151674 inferred by scTour shown in UMAP-reduction gene expression space. (**c**) RNA velocity of sample ID151674 inferred by scTour shown in UMAP-reduction gene expression space. Due to the misaligned hierarchical organization in the low-dimensional gene space, RNA velocity shows an incorrect trajectory, moving directly from layer 6 to layer 1 and continuing its evolution from layer 1. (**d**) Sample ID151674 displayed in physical space, annotated by different layers, illustrates the true hierarchical organization of the cerebral cortex. (**e**) Pseudo-time of sample ID151674 shown in physical space. (**f**) SpaRNA velocity of sample ID151674 inferred by iSORT shown in physical space. (**g**) Spatial transcriptomics of a human developing heart at 9 post-conception week. Different colors represent different cell types. (**h**) SpaRNA velocity on the human developing heart visualized by iSORT, characterizing the sequential appearance of different cell types in the human heart during development. (**i**) SpaRNA velocity inferred by iSORT on the mouse embryo dataset. iSORT successfully predicted the spatial differentiation trajectories of different cell types, revealing the patterns of spatial development.

evaluating spatial velocity inference. Unlike other methods for inferring RNA velocity, which require paired scRNA-seq data to estimate spliced and unspliced gene expressions, iSORT infers spatial RNA velocity directly from ST data.

In this analysis, iSORT successfully predicted the spatial differentiation trajectories of different cell types, revealing distinct patterns of spatial development (Figs 6i and W in S1 Text). We observed the anterodorsal differentiation of gut tube cells and the differentiation of mesodermal cells toward splanchnic mesoderm. Meanwhile, mesodermal and ectodermal cells followed distinct differentiation trajectories with clear spatial localization patterns, which are consistent with known biological processes of embryonic development.

## Discussion

In this study, we introduce a comprehensive analysis tool for exploring and deciphering the spatial organization of cells. iSORT leverages the concept of density ratio transfer to integrate scRNA-seq data with one or more ST references and reconstruct the spatial structure of scRNA-seq data at single-cell resolution. By using the reconstruction mapping, iSORT can recover spatial patterns of gene expressions, identify SOGs that are critical in driving spatial organization, and introduce a new quantity SpaRNA velocity that captures pseudo-growth of tissue to model growth dynamics of tissue. iSORT is found to be robust under different situations, such as diverse sample sources and additional distortion of references. We also conducted sequencing experiments on human arteries with and without atherosclerosis to exhibit the practicality of iSORT in finding biomarkers and researching diseases.

Several areas of improvements can be addressed for iSORT.

First, during the training process by iSORT, ST references and scRNA-seq data are required to be the same or similar tissues, although their sampled objects, tissue shapes and cell-type distributions may be different. Allowing the training on more samples including different organs and species could significantly improve the power of the method. As large models in the single-cell domain are developing rapidly, models like scGPT [68] and scFoundation [69] demonstrate the feasibility and benefits of using large-scale pretrained models in understanding complex biological data. The iSORT framework, with its unique focus on integrating single-cell and ST data, could be ideally positioned to leverage these advancements. By incorporating elements from these large models, iSORT could enhance its capabilities in reconstructing spatial organization from diverse tissue types and conditions.

Second, although we employ a shared gene strategy during the training phase, we can first perform gene imputation on both datasets before training, thereby expanding the available gene set beyond the initial shared genes. With this expanded gene set, iSORT can be trained on a broader set of genes rather than being restricted to the initial common subset, allowing for a more comprehensive spatial reconstruction of single-cell organization. Subsequently, we can identify SOGs from this expanded set, assess their spatial dominance, and further select them as spatially dominant genes for downstream spatial analysis. This approach allows researchers to explore potential SOGs that were not initially included in the shared gene subset, thereby enhancing iSORT's utility in investigating the spatial roles of specific genes and improving the overall robustness of ST integration.

Third, cell-cell communication (CCC) inference provides a powerful way in analyzing spatial organization of tissue [70,71]. With the spatial information inferred by iSORT on cells, one may use matched scRNA-seq data and ST data for CCC inference, in particular, CCC may be scrutinized in conjunction with SOGs to analyze pattern-driven CCC.

Meanwhile, compared to traditional spot deconvolution methods, iSORT exhibits three advantages: (1) iSORT directly predicts the spatial positions of single cells in continuous

space. (2) iSORT integrates information from multiple slices. (3) iSORT infers pseudo-growth trajectories using SpaRNA velocity.

Traditional spot deconvolution methods typically assign single cells to spots or integrate data to infer the cell type proportions within spots. Although these methods achieve high accuracy, they are limited to analyzing a single ST slice and cannot utilize information from multiple slices. iSORT's multi-slice training approach overcomes this limitation. One aspect is that integrating data from multiple slices significantly improves the accuracy of spatial reconstruction. Another is that incorporating slices in different states enables iSORT effectively identify key genes that determine spatial structures. This capability makes iSORT uniquely advantageous in disease research—leveraging spatial information from multiple ST slices to identify SOGs, thus providing a method for detecting pathogenic genes. Besides the atherosclerosis described in this study, iSORT could offer a new perspective for revealing broader disease-related pathways.

Moreover, iSORT's mapping-based approach allows it to incorporate the concept of RNA velocity to infer pseudo-growth trajectories on ST data, thereby capturing dynamic changes within tissues. By inferring SpaRNA velocity, iSORT not only reflects the organizational and differentiation processes within tissues, but also reveals spatiotemporal dynamics. More recently, SIRV [72] was developed to infer RNA velocity in the physical space of ST by integrating scRNA-seq and ST data. Specifically, it utilizes splicing information from scRNA-seq to predict spliced and unspliced RNA levels in the ST data, enabling RNA velocity inference within the tissue structure. In contrast, we propose a quantity named SpaRNA velocity. Instead of relying on external sequencing references, SpaRNA velocity is computed solely from ST data and projected onto the ST slice, providing pseudo-growth trajectories that model how cells transition from their progenitors in space. Other methods of RNA velocity, such as scVelo [73], can also be appropriately modified for SpaRNA velocity inference. Besides, the inference of pseudo-time is another important aspect in assessing cell developmental status [74,75]. Inference of SpaRNA velocity directly from ST data provides a promising framework for studying pseudo-time and could serve as a meaningful topic for future research.

Recently, several methods have been developed for joint 3D reconstruction of biological tissues using multiple 2D slices [76]. The framework of iSORT, in principle, can be generalized to incorporate 3D spatial reconstruction and the pseudo-growth dynamics.

In summary, iSORT provides a promising computational framework for analyzing and integrating scRNA-seq and ST data, offering perspectives in analyzing patterns and organization of spatial tissues.

## Materials and methods

### Collection of human artery samples and ethics statement

The human artery data were collected from patients undergoing coronary artery bypass grafting (CABG) or heart transplantation at Zhongshan Hospital, Fudan University. Written informed consent was obtained from each participant before surgery, with the study approved by the Ethics Committee of Zhongshan Hospital, Fudan University (ethical approval number: B2022-031R), and conducted in strict accordance with the principles outlined in the Declaration of Helsinki.

For scRNA-seq, artery samples were processed to generate single-cell suspensions. The process involved cell separation, labeling, and library construction, followed by sequencing on the Illumina NovaSeq platform. The sequencing data were processed with Cell Ranger (10x

Genomics, version cellranger-6.0.0), aligning them to the human reference genome (GRCh38) to create a matrix of gene expression barcodes.

For ST, the samples that passed quality inspection were re-sectioned for permeabilization experiments. The Visium Spatial Tissue Optimization Slide & Reagent Kit (PN-1000193, 10X Genomics) was used to release mRNA from cells and bind it to spatially barcoded oligonucleotides on the slides. Imaging to determine the appropriate permeabilization time was performed using Leica Aperio CS2 and Leica THUNDER Imager Tissue. The libraries were then prepared using the Visium Spatial Gene Expression kit (PN-1000184, 10X Genomics).

To compare the structural and morphological differences between normal vessels and those affected by atherosclerosis, tissue sections prepared for ST were additionally stained with hematoxylin and eosin, and subsequently analyzed using a Leica Aperio CS2.

## Data preprocessing

First, the raw gene expression for each scRNA-seq cell or ST spot is log-transformed and normalized by

$$x_{ij} = \ln\left(s \cdot \frac{d_{ij}}{\sum_{k=1}^{n} d_{kj}} + 1\right), \tag{1}$$

where $d_{ij}$ is the raw expression of gene $i$ in cell/spot $j$, $n$ is the total number of genes, $s$ is a scaling factor with default value 10000, and $x_{ij}$ is the normalized gene expression. Then, $x_{ij}$ is scaled by $z$-score

$$x_{ij} \leftarrow \frac{x_{ij} - \mu_i}{\sigma_i}, \tag{2}$$

where $\mu_i$ and $\sigma_i$ are the mean value and standard variation of gene $i$ across cells/spots, respectively. Next, highly variable genes are selected separately from the scRNA-seq and ST datasets. Then, we select the common subset of highly variable genes between the two datasets as the genes for subsequent tasks. All these preprocessing procedures can be realized by Python codes or packages such as Seurat [38] and Scanpy [77].

## iSORT framework

In this section, we describe the framework of iSORT for the single ST reference case. Details for the case with multiple ST references can be found in Note A in S1 Text.

**Spatial organization mapping.** Denote $X_{sc} = \{x_{sc_1}, x_{sc_2}, ..., x_{sc_M}\}$ as the expressions for the scRNA-seq data, and $X_{st} = \{x_{st_1}, x_{st_2}, ..., x_{st_m}\}$ for the ST reference, where $M$ and $m$ are the sample sizes. The location of ST data is $Y_{st} = \{y_{st_1}, y_{st_2}, ..., y_{st_m}\}$. Suppose that there are $H$ highly variable genes after preprocessing, then we have $x_{sc_i} \in \mathbb{R}^H$, $x_{st_j} \in \mathbb{R}^H$, $y_{st_j} \in \mathbb{R}^2$, $i = 1, 2, ..., M$, $j = 1, 2, ..., m$. Let $p_{sc}(x, y)$ and $p_{st}(x, y)$ be the joint probability density functions (pdfs) of the scRNA-seq data and the ST data, respectively. Due to the variety in samples and discrepancies in technologies, the marginal cell-type distributions and expression scales are usually different between $X_{sc}$ and $X_{st}$, i.e.

$$p_{st}(x) \neq p_{sc}(x), \quad \text{and} \quad p_{st}(y|x) \neq p_{sc}(y|x). \tag{3}$$

In iSORT, we employ the reference-based co-embedding approach as Seurat [38] does. $X_{sc}$ serves as the query and $X_{st}$ serves as the reference. Denote the variables after the co-embedding as $Z_{sc}$ and $Z_{st}$, respectively. Here, the function $h$ maps the gene expression space

to the latent space, represented by $z = h(x)$. Under the postulation that a cell's spatial location is intrinsically related to its latent expression, we obtain

$$p_{st}(z) \neq p_{sc}(z), \quad \text{and} \quad p_{st}(y|z) = p_{sc}(y|z), \tag{4}$$

which implies that the samples with similar features $z$ have close spatial coordinates $y$. We represent the mapping between latent expressions and locations as $y = g(z)$, where $g : \mathbb{R}^H \rightarrow \mathbb{R}^2$. The task is to give an estimation of $g$ and determine the location of $X_{sc}$ by $f(X_{sc}) := g \circ h(X_{sc})$. Several studies have discussed the construction of $g$ considering the covariance bias [78,79]. iSORT addresses the estimation as a learning task to minimize a loss function

$$\mathbb{E}_{(z,y) \sim p_{sc}} \left[ L(z, y; g) \right] = \mathbb{E}_{(z,y) \sim p_{st}} \left[ \frac{p_{sc}(z, y)}{p_{st}(z, y)} L(z, y; g) \right] \tag{5}$$

$$= \mathbb{E}_{(z,y) \sim p_{st}} \left[ \frac{p_{sc}(z) p_{sc}(y|z)}{p_{st}(z) p_{st}(y|z)} L(z, y; g) \right]$$

$$= \mathbb{E}_{(z,y) \sim p_{st}} \left[ \frac{p_{sc}(z)}{p_{st}(z)} L(z, y; g) \right], \tag{6}$$

where Eq (4) is applied in the last step, and $L(\cdot, \cdot; g)$ is the loss function. For the spatial reconstruction, $L$ is usually selected as the mean squared error loss, i.e. $L(z, y; g) = \|g(z) - y\|_2$. The density ratio $w(z) = p_{sc}(z)/p_{st}(z)$ is the key to address the different cell-type distributions between the scRNA-seq and ST data.

For the $m$ samples in the ST slice, the minimization of (6) is discretized as

$$\underset{g}{\text{argmin}} \frac{1}{m} \sum_{j=1}^{m} w_j L(z_{st_j}, y_{st_j}; g) + \Omega(g), \tag{7}$$

where $w_j = w(z_{st_j})$ is the sample-specific density ratio and $\Omega(g)$ is the normalization term. Once $w_j$s are known, we can obtain $g$ by optimizing Eq (7) and then apply $f$ on $X_{sc}$.

**Estimation of the density ratio.** iSORT estimates the density ratio $w(z) = p_{sc}(z)/p_{st}(z)$ by the method of KLIEP [79]. KLIEP demonstrates computational efficiency, stability in performance, and effective mitigation of overfitting. Specifically, $w(z)$ is represented linearly by a set of basis functions, i.e.

$$\hat{w}(z) = \sum_{i=1}^{k} a_i \phi_i(z), \tag{8}$$

where $\hat{w}$ is the approximation of $w$, $\{\phi_i\}_{i=1}^{k}$ are $k$ given basis functions satisfying $\phi_i(z) \geq 0$, and $a_i$s are the coefficients to be determined. By default, Gaussian basis functions (GBFs) are selected as $\phi_i$, offering qualities of smoothness, locality, and universal approximation capability [80].

KLIEP estimates $\{a_i\}_{i=1}^{k}$ by minimizing the KL divergence between $p_{sc}(z)$ and $\hat{p}_{sc}(z) = \hat{w}(z) p_{st}(z)$, i.e.

$$\text{KL}[p_{sc}(z) \| \hat{p}_{sc}(z)] = \int p_{sc}(z) \ln\left( \frac{p_{sc}(z)}{\hat{w}(z) p_{st}(z)} \right) dz$$

$$= \int p_{sc}(z) \ln\left( \frac{p_{sc}(z)}{p_{st}(z)} \right) dz - \int p_{sc}(z) \ln \hat{w}(z) dz. \tag{9}$$

Only the second term in Eq (9) contains $a_i$, and the optimization is equivalent to maximize

$$J := \int p_{\text{sc}}(\boldsymbol{z}) \ln \hat{w}(\boldsymbol{z}) d\boldsymbol{z} \approx \frac{1}{M} \sum_{j=1}^{M} \ln \left( \sum_{i=1}^{k} a_i \phi_i(\boldsymbol{z}_{\text{sc}_j}) \right), \tag{10}$$

where $\boldsymbol{z}_{\text{sc}_j}$s are the latent expressions of scRNA-seq data, and $M$ is the number of scRNA-seq samples. Considering the constraint of pdf as

$$\int \hat{p}_{\text{sc}}(\boldsymbol{z}) d\boldsymbol{z} = \int \hat{w}(\boldsymbol{z}) p_{\text{st}}(\boldsymbol{z}) d\boldsymbol{z} = 1, \tag{11}$$

$\{a_i\}_{i=1}^{k}$ is finally estimated by solving

$$\begin{aligned} \underset{\{a_i\}}{\operatorname{argmax}} \quad & \sum_{j=1}^{M} \ln \left( \sum_{i=1}^{k} a_i \phi_i(\boldsymbol{z}_{\text{sc}_j}) \right) \\ \text{s.t.} \quad & \frac{1}{m} \sum_{j=1}^{m} \sum_{i=1}^{k} a_i \phi_i(\boldsymbol{z}_{\text{st}_j}) = 1, \\ & a_i \geq 0, \ i = 1, 2, \dots, k. \end{aligned} \tag{12}$$

**Optimizing the reconstruction mapping.** Neural network technology can uncover the nonlinear mapping between variables with just a few hidden layers [81,82]. To solve the optimization problem (7) and approximate spatial organization mapping $\boldsymbol{f}$, a multi-layer BP neural network was applied (Fig 1b). Dropout is used to relieve the overfitting. Detailed architecture is described in the SI.

## Identification of SOGs

SOGs are the genes most relevant to the spatial organization of tissues. Within the framework of iSORT, $\boldsymbol{f}$ maps the gene expression data $\boldsymbol{x} \in \mathbb{R}^n$ to a two-dimensional spatial coordinate $\boldsymbol{y} \in \mathbb{R}^2$, i.e. $\boldsymbol{y} = \boldsymbol{f}(\boldsymbol{x})$. We proposed the SOG index of gene $g$ as

$$I_g := \|\partial_{x_g} \boldsymbol{f}(\boldsymbol{x})\| = \mathbb{E} \left\{ \frac{1}{2} \sum_{p=1}^{2} \left| \frac{\partial y_p}{\partial x_g} \right| \right\}, \tag{13}$$

where $y_1, y_2$ are the components of $\boldsymbol{y}$, $x_g$ is the gene $g$ expression in $\boldsymbol{x}$, and the expectation $\mathbb{E}$ is taken over all cells/samples. SOGs are identified as the genes with top $I_g$ scores, reflecting their significant contributions to the spatial configuration and biological functionality.

## Inference of SpaRNA velocity

iSORT proposes a way named SpaRNA velocity to visualize the spatial growth of tissues. For a cell with gene expression $\boldsymbol{x}$, we suppose that its RNA velocity $\boldsymbol{v}_{\text{RNA}}$ is obtained in the gene expression space by scTour [37] or other algorithms. Using the mapping $\boldsymbol{f}$, we can define the SpaRNA velocity as

$$\boldsymbol{v}_{\text{st}} = \boldsymbol{f}(\boldsymbol{x} + \boldsymbol{v}_{\text{RNA}}) - \boldsymbol{f}(\boldsymbol{x}). \tag{14}$$

The SpaRNA velocity allows us to further explore the dynamics of tissue growth in physical space, and provides a comprehensive view of the cellular organization.

## Simulation of the coarse-grained ST data

FISH technology can provide high-resolution ST data [83] at single-cell scale, while most sequence-based approaches such as 10X Visium can only achieve spot-resolution. The diameter of a spot in 10X Visium is 55 μm, while the distance between the centers of two adjacent spots is 100 μm. One spot may contain 1–10 cells.

To test the effectiveness of iSORT when using low-resolution ST data as the reference, we simulate the coarse-grained ST from the seqFISH data in mouse embryo and FISH data in *Drosophila* embryo experiments. In the simulation, we initially divided the area into a uniform grid, with the intersections of the lines serving as spots. The radius of each coarse spot is set to be one quarter of the adjacent spot spacing, and the gene expression is set as the accumulated value of all cells within the radius. We only considered spots with gene expression greater than zero.

## Incorporating variable noise intensities into semi-simulated data

To evaluate the robustness of iSORT under varying noise conditions, we incorporated Gaussian noise into the single-cell gene expression data. The noise was added at a gene-specific level, with the intensity controlled by a noise ratio parameter $\sigma$, which adjusts the proportion of noise relative to the intrinsic variability of each gene. This process simulates realistic noise levels in scRNA-seq data and assesses the impact of noise on iSORT's ability to reconstruct spatial patterns.

Let $X_{ij}$ be the expression value of gene $j$ in cell $i$, and $\text{Var}(X_j)$ the intrinsic variance of gene $j$ across cells. The Gaussian noise for each gene is generated as:

$$\mathcal{N}_{ij} \sim \mathcal{N}(0, \sqrt{\sigma \cdot \text{Var}(X_j)})$$

where $\mathcal{N}_{ij}$ represents the noise added to $X_{ij}$. The noise ratio $\sigma$ determines the noise intensity, with higher values corresponding to more noise. For instance, $\sigma = 0.1$ introduces 10% noise variance relative to the gene's intrinsic variance.

The noisy gene expression matrix $X'$ is then computed as:

$$X'_{ij} = \max(X_{ij} + \mathcal{N}_{ij}, 0)$$

ensuring non-negative values for the final expression data.

## Toy model of spatial transcriptomics

To simulate ST data and distinguish between SOG and SVG, we developed a toy model that generates gene expression and spatial coordinates with spatial dependencies.

The gene expression matrix $X \in \mathbb{R}^{n \times m}$ was created, where $X_{ij}$ represents the expression of gene $j$ in the sample $i$. The expression values for the first four genes were sampled from a uniform distribution $U(0, 2)$, and the spatial coordinates were generated as:

$$y_{i,1} = 2 \cdot \sin(X_{i,1}) + 1.5 \cdot X_{i,2}^2 + \mathcal{N}(0, R)$$

$$y_{i,2} = -\exp(-X_{i,3}) + 2.5 \cdot \cos(X_{i,4}) + \mathcal{N}(0, R)$$

where $\mathcal{N}(0, R)$ is Gaussian noise and $R$ is set to 0.1.

To simulate spatially dependent genes, we introduced a spatial weight matrix based on the Euclidean distance $d_{ij}$ between spatial coordinates, with spatial correlation controlled by the parameter $W$:

$$w_{ij} = \exp\left(-\frac{d_{ij}}{W}\right)$$

In our simulation, $W$ was set to 0.6. Gene expression values for the remaining genes were generated as:

$$X_i = b + \sum_j w_{ij} \cdot \mathcal{N}(0,1)$$

This ensures high spatial autocorrelation and a strong Moran's I index, with the first four genes determining spatial positions and the remaining six genes exhibiting high spatial correlation.

## Supporting information

**S1 Text. Supplementary information.** The supplementary document provides eight notes, ten tables, and twenty-three supplementary figures for the main text.
(PDF)

## Author contributions

**Conceptualization:** Yecheng Tan, Ai Wang, Qing Nie, Jifan Shi.

**Formal analysis:** Yecheng Tan, Ai Wang, Jifan Shi.

**Methodology:** Yecheng Tan, Jifan Shi.

**Software:** Yecheng Tan, Jifan Shi.

**Supervision:** Wei Lin, Yan Yan, Qing Nie, Jifan Shi.

**Visualization:** Yecheng Tan, Zezhou Wang, Jifan Shi.

**Writing – original draft:** Yecheng Tan, Jifan Shi.

**Writing – review & editing:** Yecheng Tan, Ai Wang, Zezhou Wang, Qing Nie, Jifan Shi.

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
