## [Decision Letter · Decision Letter 0]

18 Dec 2024

PCOMPBIOL-D-24-01905

Transfer learning of multicellular organization via single-cell and spatial transcriptomics

PLOS Computational Biology

Dear Dr. Tan,

Thank you for submitting your manuscript to PLOS Computational Biology. After careful consideration, we feel that it has merit but does not fully meet PLOS Computational Biology's publication criteria as it currently stands. Therefore, we invite you to submit a revised version of the manuscript that addresses the points raised during the review process.

Please submit your revised manuscript within 60 days Feb 17 2025 11:59PM. If you will need more time than this to complete your revisions, please reply to this message or contact the journal office at ploscompbiol@plos.org. Please include the following items when submitting your revised manuscript:

We look forward to receiving your revised manuscript.

Kind regards,

Jean Fan

Academic Editor

PLOS Computational Biology

Sushmita Roy

Section Editor

PLOS Computational Biology

**Journal Requirements:**

3) Please ensure that the funders and grant numbers match between the Financial Disclosure field and the Funding Information tab in your submission form. Note that the funders must be provided in the same order in both places as well. State the initials, alongside each funding source, of each author to receive each grant. For example: "This work was supported by the National Institutes of Health (####### to AM; ###### to CJ) and the National Science Foundation (###### to AM)." State what role the funders took in the study. If the funders had no role in your study, please state: "The funders had no role in study design, data collection and analysis, decision to publish, or preparation of the manuscript.".

**Reviewers' comments:**

Reviewer's Responses to Questions

**Comments to the Authors:**

Reviewer #1: This manuscript presents a transfer learning framework for predicting the spatial positions of single cells based on scRNA-seq and spatial transcriptomics data. While the topic is interesting, the benchmark is conducted with low standards, with only mapping-based algorithms compared. In fact, many deconvolution-based algorithms have been proposed to solve the same questions, including RCTD, cell2location, DestVI, CARD, Redeconve, CytoSPACE, etc. Deconvolution-based algorithms have higher accuracy than mapping-based algorithms. Without demonstrating the advantages compared with currently available deconvolution-based algorithms, the current tool cannot be recommended for publication.

Reviewer #2: In this study, Shi et al. present iSORT, an algorithm that combines single-cell data and spatial transcriptomics through a transfer learning approach to reconstruct spatial organization. The authors benchmark iSORT against state-of-the-art methods using datasets from the human dorsolateral prefrontal cortex (DLPFC), mouse embryo, and mouse brain, assessing its performance in spatial distribution reconstruction. iSORT effectively identifies spatial patterns at the single-cell level, pinpoints spatial-organizing genes (SOGs) that drive these patterns, and infers pseudo-growth trajectories using the concept of SpaRNA velocity. Overall, iSORT demonstrates accuracy and efficiency in mapping single-cell and spatial transcriptomics data.

However, the authors should address several issues related to this methods.

1. In this study, the most scRNA-seq input was derived from the gene expression of spatial transcriptomics (ST) data at each spot, with the spatial coordinates removed. Since Visium data does not achieve true single-cell resolution and SeqFish only detects thousands of genes, these cannot be considered genuine single-cell data. It is crucial to evaluate iSORT's performance on different ST platform paired with real single cell data to determine whether it can effectively address batch effects between single-cell and ST technologies, enabling the reconstruction of spatial patterns and facilitating downstream analyses.

2. The manuscript compares iSORT with other methods using simulated SC data. However, additional comparative analyses are necessary to demonstrate the robustness of iSORT's performance. The authors analyze single-cell data generated from spatial transcriptomics data, but this method may not accurately represent real-world situations where SC and ST data originate from different samples and different technologies. It could be beneficial for the authors to evaluate iSORT's performance by incorporating various noise levels into the SC data.

3. When the single-cell data and spatial transcriptomics data have different numbers of genes, how should genes be selected for integration, particularly when the ST data, obtained from image-based technologies, includes only a few hundred genes? Can iSORT assist in identifying more spatially dominant genes from the scRNA-seq data, which contains a larger gene set?

4. The authors claim that the SOGs identified by iSORT are more effective than the SVGs detected by Moran's I and SpatialDE based on gene knockout experiments. What are the key differences genes between the SOGs and SVGs? The authors should provide additional evidence, such as gene expression patterns for the SOGs, and discuss the biological significance of these genes and their relationship to tissue structure and domains.

5. The results in Fig. 5 is confusing. It employs UMAP reduction to present the pseudo-time and RNA velocity results of scTour, while iSORT's results are shown in physical space. It might be better to use a consistent format to demonstrate and compare the two different methods.

Reviewer #3: In this manuscript, the authors established iSORT, a novel mapping method between single-cell and spatial transcriptomes. They also proposed two downstream analysis methods for spatial RNA velocity estimation and spatial gene identification, respectively. I have the following concerns:

1. iSORT is a reference-based mapping method. The authors should compare iSORT with state-of-the-art reference-based mapping methods such as CytoSPACE (https://www.nature.com/articles/s41587-023-01697-9), STEM (https://www.nature.com/articles/s42003-023-05640-1) and Celloc (https://onlinelibrary.wiley.com/doi/10.1002/smsc.202400139).

2. Inclusion of multiple ST references is a highlight feature of iSORT, but the tests demonstrating such advantage are not sufficient.

3. Although the authors emphasize the introduction of transfer learning in their algorithm, I could not exactly find which part of the algorithm adopts such a transfer learning strategy in Figure 1. It seems that the term "density ratio transfer" would more accurately describe the strategy.

4. For spatial RNA velocity, the authors could compare their method with the latest method SIRV (https://academic.oup.com/nargab/article/6/3/lqae100/7728020).

5. For spatial gene identification, the rationale for equation (13) is not clear. In addition, the authors could perform ssGSEA/GSVA analysis to check whether the spatial-organizing genes found by their method are more enriched for known key development-related genes or disease genes.

6. To better quantify the performance, the authors could compare their spatial-organizing gene identification method with the previous methods in simulated datasets, using a benchmarking strategy similar to the recent report (https://genomebiology.biomedcentral.com/articles/10.1186/s13059-023-03145-y).

**Have the authors made all data and (if applicable) computational code underlying the findings in their manuscript fully available?**

Reviewer #1: Yes

Reviewer #2: Yes

Reviewer #3: Yes

PLOS authors have the option to publish the peer review history of their article (what does this mean?). If published, this will include your full peer review and any attached files.

Reviewer #1: No

Reviewer #2: No

Reviewer #3: No

**Figure resubmission:**
---

## [Decision Letter · Decision Letter 1]

24 Mar 2025

Dear Mr Tan,

We are pleased to inform you that your manuscript 'Transfer learning of multicellular organization via single-cell and spatial transcriptomics' has been provisionally accepted for publication in PLOS Computational Biology.

Best regards,

Jean Fan

Academic Editor

PLOS Computational Biology

Sushmita Roy

Section Editor

PLOS Computational Biology

Reviewer's Responses to Questions

**Comments to the Authors:**

Reviewer #1: The authors have addressed all my concerns.

Reviewer #2: The authors have addressed all my comments. Thank authors for their efforts.

Reviewer #3: The authors have largely addressed my previous concern, making the overall quality of the manuscript suitable to be accepted.

**Have the authors made all data and (if applicable) computational code underlying the findings in their manuscript fully available?**

Reviewer #1: None

Reviewer #2: Yes

Reviewer #3: Yes

PLOS authors have the option to publish the peer review history of their article (what does this mean?). If published, this will include your full peer review and any attached files.

Reviewer #1: No

Reviewer #2: **Yes: **Can Yang

Reviewer #3: No

---

## [Editor Report · Acceptance letter]

PCOMPBIOL-D-24-01905R1

Transfer learning of multicellular organization via single-cell and spatial transcriptomics

Dear Dr Tan,

I am pleased to inform you that your manuscript has been formally accepted for publication in PLOS Computational Biology. Your manuscript is now with our production department and you will be notified of the publication date in due course.

With kind regards,

Anita Estes
